# Hydrogen Separation and Purification from Various Gas Mixtures by Means of Electrochemical Membrane Technology in the Temperature Range 100–160 °C

**DOI:** 10.3390/membranes11040282

**Published:** 2021-04-10

**Authors:** Leandri Vermaak, Hein W. J. P. Neomagus, Dmitri G. Bessarabov

**Affiliations:** 1HySA Infrastructure Centre of Competence, Faculty of Engineering, Potchefstroom Campus, North-West University, Potchefstroom 2520, South Africa; 2Centre of Excellence in Carbon Based Fuels, Faculty of Engineering, School of Chemical and Minerals Engineering, Potchefstroom Campus, North-West University, Potchefstroom 2520, South Africa; hein.neomagus@nwu.ac.za

**Keywords:** electrochemical hydrogen separation/purification, high-temperature proton exchange membrane (PEM), polarization curve

## Abstract

This paper reports on an experimental evaluation of the hydrogen separation performance in a proton exchange membrane system with Pt-Co/C as the anode electrocatalyst. The recovery of hydrogen from H_2_/CO_2_, H_2_/CH_4_, and H_2_/NH_3_ gas mixtures were determined in the temperature range of 100–160 °C. The effects of both the impurity concentration and cell temperature on the separation performance of the cell and membrane were further examined. The electrochemical properties and performance of the cell were determined by means of polarization curves, limiting current density, open-circuit voltage, hydrogen permeability, hydrogen selectivity, hydrogen purity, and cell efficiencies (current, voltage, and power efficiencies) as performance parameters. High purity hydrogen (>99.9%) was obtained from a low purity feed (20% H_2_) after hydrogen was separated from H_2_/CH_4_ mixtures. Hydrogen purities of 98–99.5% and 96–99.5% were achieved for 10% and 50% CO_2_ in the feed, respectively. Moreover, the use of proton exchange membranes for electrochemical hydrogen separation was unsuccessful in separating hydrogen-rich streams containing NH_3_; the membrane underwent irreversible damage.

## 1. Introduction

Hydrogen shows great potential regarding future energy prospects, especially in the potential low-carbon energy system known as ‘the hydrogen economy’. Some of the benefits that promote hydrogen as an energy carrier include the following. Hydrogen is the most abundant element in the universe; it is the lightest element with the highest energy content among all the existing fuels [1,2]; it is a highly efficient and clean energy carrier [3], with only water as a by-product at conversion to energy [2,3]; and, furthermore, hydrogen lends itself to a variety of storage methods (e.g., gaseous, liquid [2,4]). Hydrogen could increase renewable energy usage, to a large extent, due to its ability to allow flexible storage of intermittent renewable energy [5]. For this reason, hydrogen is seen as a key solution to the global climate change [2], with the global ambition to maintain the increase in the average global temperature to <2 °C above pre-industrial levels, as stated by the Paris Agreement of 2015 [6].

Among other technologies that use hydrogen, specifically for mobility applications, proton exchange membrane (PEM) fuel cells (FCs) are attracting the most attention due to their favorable operational characteristics, such as quiet operation, near-zero emissions, high power densities, high conversion efficiencies, etc. [7]. However, for effective operation, high-purity hydrogen (>99.97%) is required [8].

Currently, large-scale hydrogen production is commonly carried out by means of steam methane reforming, coal gasification, or partial oxidation of hydrocarbons [9,10,11]. Although not widely implemented, there are several small-scale hydrogen production methods, including partial oxidation of biomass, electrolysis and auto-thermal processes, and small-scale steam methanol reformers [9,12]. However, both large- and small-scale hydrogen production methods produce hydrogen-rich streams commonly accompanied by impurities such CO_2_, CO, N_2_, and CH_4_, among others [13,14,15,16,17] (e.g., steam methane reforming (CO, CO_2_, CH_4_), dry shifted syngas (N_2_, CO, CO_2_, CH_4_, H_2_S), hot shifted syngas (N_2_, CO, H_2_S) [18,19]).

There are now several technological approaches being used to extract and purify hydrogen from gas mixtures, utilizing various characteristics of hydrogen under different industrial conditions. Common approaches for hydrogen recovery include adsorbing the impurities (pressure swing adsorption (PSA)), condensing the impurities (cryogenic distillation) or by using hydrogen-selective membranes [13,16,20,21]. See Table 1.

Although PSA and cryogenic distillation technologies are both commercial processes, multiple units are required and, in some instances, supplementary wash columns are required to remove CO and CO_2_ [13]. Pressure-driven membrane processes are considered to be better candidates for hydrogen production because they are not as energy intensive and they yield high-purity hydrogen [25]. However, even though membrane technologies are advantageous over the other purification methods mentioned, they commonly depend on high-pressure feed streams, and hydrogen embrittlement is often experienced [13].

As an alternative to pressure-driven membranes, electrochemical membrane technology based on PEMs has been utilized for hydrogen separation and it has been used to effectively recover hydrogen from H_2_/N_2_, H_2_/CO_2_/CO, and H_2_/CH_4_ hydrogen-rich gas mixtures [26]. Electrochemical hydrogen separation (EHS) offers several advantages over its competitors [27,28], including high-purity hydrogen from one-step operation [29], simultaneous hydrogen compression and separation is possible, as well as hydrogen separation is achieved at low cell voltages with a high separation efficiency [30]. An additional benefit of this electrochemical method is that CO_2_, if present in the feed, is concentrated, and it can be captured and stored without the requirement for any further treatment, thus, reducing greenhouse gas emissions [26,30].

Perfluorinated sulfonic acid (PFSA) membranes (mainly Nafion-based), typically operated at 60–80 °C, are mostly used as PEMs due to their high proton conductivity, mechanical integrity, and chemical stability [13]. However, one major challenge of Nafion membranes include the complex water management of the membrane [13]. Furthermore, swelling-induced stresses could result from water adsorption of the membrane [31]. Another major disadvantage is deactivation of the electrocatalyst (mainly based on Pt) when small amounts of CO (10–1000 ppm) are present [13,14], which results in a significant performance drop of the electrochemical cell [14].

High-temperature (HT)-PEMs (operating temperatures: 100–200 °C) doped with phosphoric acid (PA) have been developed to overcome the limitations associated with low-temperature (LT) PSFA membranes [32,33,34]. HT PA-doped PEMs conduct protons well, even in anhydrous conditions, which eliminates the requirement of pre-humidifying the feed stream [35]. Some other advantages include their fast electrode kinetics, enhanced mass transport, higher tolerance to impurities such as CO (up to 3 vol.%), and better sustainability [13,36,37,38]. Recent advances in other HT membranes, specifically for fuel cells, have been made [39,40,41]. These include, for example, phosphonated poly(pentafluorostyrene) for fuel cells and fuel cells based on quaternary ammonium-biphosphate ion pairs. A comparison of the properties of typical HT and LT PEMs can be found in Reference [32]. In terms of EHS, limited information is currently available on HT PEMs separation compared to LT separation (Nafion membranes) [42].

In the present work, hydrogen was separated from various concentrations of H_2_/CH_4_, H_2_/CO_2_ and H_2_/NH_3_ mixtures at higher temperatures of 120–160 °C. The gas mixtures were chosen based on their industrial relevance in processes related to syngas, the water–gas shift reaction, the power-to-gas (P2G) concept, and ammonia production. A HT-PEM (TPS^®^-based) was selected as the commercially available membrane choice due to the higher operating temperatures of these membranes compared to the PBI membranes of the same supplier (Advent technologies, Inc., Cambridge, MA, USA). To understand, and evaluate, the characteristics of the membrane, polarization curves and electrochemical impedance spectroscopy (EIS) were used. An in-line gas chromatograph (GC) was used to evaluate the separation performance of the membrane. This was done by measuring the composition of the permeate stream and comparing it with the composition of the feed stream, using the hydrogen selectivity parameter.

## 2. Working Principle

An electrochemical hydrogen separator (EHS), illustrated in Figure 1, is principally based on the following mechanism: a molecular gaseous hydrogen-containing gas stream is fed, at low pressure, at the anode side of the PEM electrochemical cell. When the hydrogen makes contact with the anode electrode, hydrogen is oxidized into protons and electrons, facilitated by a Pt-based catalyst. Hydrogen is then selectively transported, in the form of protons, through the PEM to the cathode compartment and the electrons then complete an electrical circuit. Finally, the protons and electrons are combined at the cathode compartment to form gaseous hydrogen. This reaction can be carried out at high pressure to improve the specific volumetric energy density of hydrogen, for storage purposes.

The hydrogen oxidation reaction does not occur spontaneously and an external load (DC power source) is required to drive the electrochemical reaction (electrolytic mode) [44]. However, minimal power is required to operate the cell, as the oxidation and reduction reactions of hydrogen are facile and nearly Nernstian/Faradic in their electrochemical behavior. Unlike conventional membrane purification systems that rely on pressure or concentration differentials to bring about separation, an EHS generates hydrogen at a rate dependent on the applied current, I (A). This is defined by Faraday’s law:(1)$$\dot{m}=\frac{MI}{nF}$$
where *ṁ* depicts the hydrogen flow rate, normally in g s^−1^ or g min^−1^, *n* is the number of electrons, *F* depicts the Faraday constant (96,485 C mol^−1^) and *M* is the molecular weight (g mol^−1^).

## 3. Materials and Methods

### 3.1. Experimental Set-Up: Cell and Membrane Electrode Assembly

Experiments were carried out using an off-the-shelf electrochemical cell with a square active area of 25 cm^2^ (Fuel Cell Technologies, Inc., Albuquerque, NM, USA). See Figure 2. The cell consisted of a catalyst coated proton-conducting membrane, which was placed between two pieces of carbon cloth, acting as gas diffusion layers (GDLs), and Ti flow fields. The cell was sealed with HT PFA gaskets (used to compress the membrane electrode assembly (MEA)). The cell was assembled with aluminium back plates and compressed using a torque wrench to ensure a uniform pressure distribution over the compressed MEA and to minimize the electrochemical and thermal resistance of the GDLs. The assembly torque for the 8 bolts was 6 Nm. The MEA consisted of carbon supported platinum (Pt/C) as the cathode catalyst and a bimetallic carbon supported platinum-cobalt (Pt-Co/C) as the anode catalyst, and a TPS^®^-based PEM (supplied by Advent Technologies, Inc., Cambridge, MA, USA). The total catalyst loading was 1.8 mg cm^−2^ (specified by Advent Technologies, Inc., Cambridge, MA, USA).

The properties of the TPS-based membranes used in the experiments can be found in Table 2.

An external DC power supply was connected to the bus plates of the electrochemical cell and the system was operated in constant current mode. The current and voltage were recorded with an electronic load, attached to a computer, making use of LabVIEW software. The cell temperature was controlled by two FIREROD^®^ (Watlow, St. Louis, MO, USA) heating rods, one at the anode side and one at the cathode side, with incorporation of a K-type thermocouple (anode side) and PID controller (DCL-33A). Furthermore, the cell was enclosed in a heat insulation jacket to reduce heat losses. The temperature at the centre of the anode back plate was taken as the cell temperature. The permeate volumetric flow rate was measured using a soap flow meter and the composition of the permeate was determined with an in-line gas chromatograph (GC) (SRI 8610C-GC; SRI Instruments, Torrance, CA, USA). See GC and bubble flow meter set-up in Figure 3. 

### 3.2. Characterization Methods Used

#### 3.2.1. Polarization Curve Measurements

The hydrogen flow rate was kept constant at 100 mL_n_/min throughout all the experiments (as described below). Polarization curves were drawn, starting at 0 A cm^−^^2^, and then increasing the current density with a step change of 0.02 A cm^−^^2^ every 10 min. The voltage output for each current density setting was recorded every 5 Sections (performed in constant current mode). This was done until a limiting current was reached or until the voltage limit (1.1 V) was reached, after which the run was terminated. The volumetric flow rate of the permeate was recorded for all current density values, making use of a soap flow meter, in order to determine the current efficiency.

#### 3.2.2. Electrochemical Impedance Spectroscopy Measurements

To determine the resistance of the MEA, in situ EIS was performed in constant current/galvanostatic mode (1 A), frequency range 100 kHz to 0.1 Hz, at 120–160 °C in hydrogen pumping mode. For feed streams containing impurities, the hydrogen flow rate was kept constant at 100 mL_n_/min H_2_. The AC current amplitude was a standard 10% of the applied direct current [14]. A Nyquist plot was generated of the impedance spectrum. 

### 3.3. Experimental Procedure

All experiments were carried out at atmospheric pressure (~86 kPa), where the cell operating temperature was varied between 120 and 160 °C. All experiments were conducted in hysteretic order as per assessment of the performance deterioration during experimentation. All the flow rates herein reported are given in normal millilitres per minute (mL_n_/min), with 0 °C and 1 bar taken as reference.

Preliminary tests performed with pure hydrogen showed that the thermal cycles (heating and cooling of the cell), which lead to the expansion and contraction of the cell components, could cause the assembly bolts to loosen. Therefore, once a thermal cycle is complete and the cell is cooled down, the assembly torque on the bolts was reinsured. This procedure was applied after each thermal cycle.

The experimental tests were organized as follow. First, as a basic characterization step, the cell was operated using pure hydrogen. The effect of temperature on the performance of the cell was investigated by varying the temperature between 120 and 160 °C, in 20 °C steps. Polarization curves and EIS graphs were drawn for data at each temperature. After the pure hydrogen tests, the separation experiments were performed. A constant hydrogen flow rate of 100 mL_n_/min was maintained throughout. The gas mixtures were tested in the following order: H_2_/CH_4_, H_2_/CO_2_, and then the H_2_/NH_3_ gas mixture, starting at the lowest impurity mixtures and moving towards the higher impurity concentrations.

## 4. Results and Discussion

### 4.1. Pure Hydrogen Experiments: Membrane Characterization

The performance of the MEA was first investigated using a pure hydrogen feed at operating temperatures between 120 and 160 °C and a hydrogen flow rate of 100 mL_n_/min. The resulting voltage–current characteristics are shown in Figure 4. It is evident that the cell performance is sensitive to small changes in temperature (20 °C). A clear increase in the cell performance was observed as the voltage values decrease, with an increase in the applied current density. This results in higher overall power efficiencies. Moreover, the limiting current density values (vertical portion of the curves) also increased with temperature. The highest achievable current density was found to be 0.40 A cm^−2^ at 160 °C.

The linear nature of the initial portion of the curves indicates that the ohmic losses (*IR*-losses) dominate the voltage losses in this region. This behavior is common in electrochemical cells and suggests that the contact resistance and the membrane resistance play a crucial role in the general performance of the cell [15,28,45]. It can therefore be concluded that the cell performance can be enhanced by minimizing both the contact and membrane resistance. In addition, the curved nature of the final portion of the curves indicates that the mass transfer limitations dominate in the high current density region. This is explained by the depletion of reactants at the surface of the electrode.

In an ideal cell, the hydrogen permeation flux is given as a function of current, according to Faraday’s law (see Equation (1)), which is derived by assuming that the hydrogen is an ideal gas. However, the real value is always smaller than that depicted by an ideal process, due to losses in the system. In this particular case, the losses can be determined by the current efficiency, which is defined as the ratio of the actual hydrogen permeation rate and the theoretical hydrogen permeation rate:(2)$$\varepsilon_{i}=\frac{{\dot{m}}_{H_{2}}\left(Experimental\right)}{{\dot{m}}_{H_{2}}\left(Theoretical\right)}$$
where ${\dot{m}}_{H_{2}}\left(Experimental\right)$ depicts actual hydrogen flow from the cathode and ${\dot{m}}_{H_{2}}\left(Theoretical\right)$ depicts the theoretical hydrogen flow rate, determined from Faraday’s law. A visual correlation between the theoretical and actual hydrogen permeation rates is plotted in Figure 5. The hydrogen permeability is also shown on the same graph (left *y*-axis). In general, the measured data show good correlation with the theoretical values.

The calculated current efficiencies, together with the voltage and power efficiencies are illustrated in Figure 6. The voltage efficiency is defined in terms of the thermal energy of the hydrogen generated (Equation (3)). This is calculated with the voltage equivalent of the heat of combustion of hydrogen (1.484 V), whereas the power efficiency can be defined as the product of both the voltage and current efficiencies (Equation (4)).
(3)$$\varepsilon_{v}=1-\frac{V}{1.484}$$
(4)$$\varepsilon_{p}=\varepsilon_{i}\times \varepsilon_{v}$$

At low current densities (<0.08 A cm^−2^) the current efficiencies were relatively low (~80%) for all temperatures considered here. However, as the current density was increased, and therefore also the hydrogen flux, the current efficiency increased to >95%. Due to the high current efficiencies observed at the higher current densities, it was concluded that the lower efficiencies at lower current densities were caused by back diffusion of hydrogen. This can be explained by the low hydrogen fluxes at low current densities. Overall, it can be concluded that the current efficiency increased with current density at constant temperature, whilst temperature seems to play a negligible role in the current efficiency. Also, the voltage efficiency increased with an increase in temperature and decreased with an increase in current density. Lastly, higher power efficiencies were reached when the temperature was increased. The maximum power efficiencies at the respective temperatures were ~91% (120 °C), ~93% (140 °C) and ~95% (160 °C) at 0.16 A cm^−2^. Therefore, 0.16 A cm^−2^ was considered the optimum operating current density with regards to the power consumption at 120–160 °C.

In order to obtain a better understanding of the temperature effect on the performance of the EHS, EIS was used to diagnose the cell. Figure 7 shows the in situ AC impedance spectrum of the EHS operated at 0.1 A cm^−2^ with variations in the temperature and the hydrogen flow rate. Similar spectra have been widely reported in the literature for both FCs [14,46] and EHSs [47,48]. The intercept of the real axis (*x*-axis) represents the ohmic resistance, which is defined as the sum of the contact resistance, electrical resistance and the proton conduction resistance in the membrane). Figure 7 shows that the impedance spectra move towards the left as the temperature is increased. The leftwards movement indicates that the ohmic resistance of the cell decreases with temperature. This is in agreement with what is reported in the literature, as the membrane can be considered as an electric insulator, which means that theoretically its resistance decreases with temperature [49].

The diameter of the impedance arc is the sum of the charge transfer and mass transfer resistance. The reduction in the arc diameter at higher temperatures indicates that there is a slight improvement in mass transport and high frequency charge transfer [49]. According to Su et al. [50], an increase in temperature enhances the cell performance due to the increase in the mass transfer rate, and a lower cell resistance, as confirmed in this work. Furthermore, the overall performance enhancement with increasing temperatures (polarization curves) can be attributed to the decrease in ohmic resistance and the improvement in mass transport and high frequency charge transfer observed in the EIS data [49].

### 4.2. Hydrogen Separation from H_2_/CH_4_ Gas Mixtures

The EHS was evaluated for the electrochemical hydrogen separation from H_2_/CH_4_ gas mixtures (10% CH_4_, 50% CH_4_, and 80% CH_4_, with the balance hydrogen), where the hydrogen flow rate was kept constant at 100 mL_n_/min. The voltage–current characteristics of the H_2_/CH_4_ mixtures at different cell temperatures and feed concentrations are shown in Figure 8 and were compared to that of pure hydrogen (refer to Figure 4).

Generally, higher voltages were reached with the H_2_/CH_4_ gas mixtures, compared to that of pure hydrogen. Here it is noteworthy that the open-current voltage (OCV), determined by the Nernst equation, is not zero, as is the case with pure hydrogen. This is due to the differences in partial pressures [13]. The OCVs were calculated for the different feed compositions and temperatures at a fixed current density of 0.1 A cm^−2^. See Table 3. These values were compared to the cell voltages corresponding to pure hydrogen at the same current density: 0.071 V (120 °C), 0.042 V (140 °C) and 0.021 (160 °C), respectively. Based on Kim et al. [47], the actual operating voltage for the gas separation can be estimated by subtracting the OCV from the measured cell voltage. However, from Table 3, it is evident that these values are still much higher than the cell voltages corresponding to pure hydrogen.

General conclusions can be drawn from the performance curves (Figure 8). Firstly, at constant inlet composition, the general performance of the cell increased with temperature. This is explained by the enhanced reaction activity (mass transfer rate) of the electrodes at higher temperatures. Similar results have been reported by Lee et al. [15]. However, at constant temperature, an increase in methane concentration leads to a decrease in the performance of the cell. This performance decrease shows that the partial pressure of the hydrogen influences the general performance of the cell. Secondly, the limiting current density values, indicating where most of the hydrogen in the H_2_/CH_4_ mixtures has been consumed and where the hydrogen partial pressure at the anode becomes small, were seen to increase with temperature and decrease with methane concentration.

The current, voltage, and power efficiencies of the H_2_/CH_4_ gas mixtures were also considered and compared to those of pure hydrogen. The experimental results obtained for 10%, 50%, and 80% CH_4_ are illustrated in Figure 9. In general, the initial current efficiencies were significantly lower than that obtained with pure hydrogen. The current efficiencies for H_2_/CH_4_ were in the range ~47–51% at 0.02 A cm^−2^, whereas those for pure hydrogen were ~70%. The current efficiencies of the H_2_/CH_4_ mixtures also increased with current density, as was the case with pure hydrogen. The voltage efficiencies, on the other hand, decreased with an increase in current density, which correlates well with the results of pure hydrogen experiments. Moreover, the voltage efficiencies seemingly increased with temperature, observed with the 50 and 80% CH_4_ inlet mixtures. However, in the case of the 10% CH_4_, this trend was only visible at current densities > 0.16 A cm^−2^.

The power efficiencies had a maximum value at 0.12 A cm^−2^ with 10% CH_4_, 0.08–0.1 A cm^−2^ with 50% CH_4_ and 0.04–0.08 A cm^−2^ with 80% CH_4_. This depicts the optimum current density for hydrogen separation from the H_2_/CH_4_ mixtures at the respective operating conditions, with regards to the power consumption. Furthermore, the power efficiencies decreased with an increase in methane concentration and increased with temperature. The highest power efficiency was, therefore, observed with the 10% CH_4_ mixture at 160 °C: ~89%. Overall, the power efficiencies obtained with the respective H_2_/CH_4_ mixtures were much lower than with pure hydrogen, especially when the hydrogen-to-methane ratio of the inlet decreased.

In terms of separation, a high purity hydrogen (>99.9%) was achieved with all the methane inlet concentrations (10%, 50%, and 80% CH_4_). This implies that a high hydrogen purity can be generated from H_2_/CH_4_ gas mixtures, in a single-stage process, regardless of the inlet methane concentration. Moreover, the hydrogen purity was slightly enhanced with an increase in temperature. Besides the hydrogen purity, the hydrogen selectivity values were also calculated and reported [51]:(5)αij=yixiyjxj
where *y_i_* and *y_j_* are the molar fractions of gas species *i* and *j* on the permeate side, while *x_i_* and *x_j_* are the molar fractions of gas species *i* and *j* on the feed side.

The relationship between the hydrogen selectivity and current density, as a function of methane concentration, is illustrated in Figure 10. At fixed temperature and impurity concentration, the hydrogen purity of the permeate stream increased with an increase in current density. This is because, according to Faraday’s law, the proton flux through the membrane increases with current density, while the impurities (in this case CH_4_) are unaffected by the current density. The permeation of the impure gas species (CH_4_) through the membrane was proportional to/dependent on the change in partial pressure. When the hydrogen partial pressure at the anode side decreased, the selectivity and the hydrogen purity of the permeate (cathode outlet) increased.

According to Rowe et al. [52], the separation performance of a membrane can be improved by reducing the temperature, as explained by the Arrhenius equation. Our findings were in agreement with this. According to Figure 10, much higher hydrogen selectivity values were achieved at lower temperatures. For the 10% methane feed stream, the hydrogen selectivity decreased from ~13,000 to ~12,750 and then to ~1980 as temperature was increased from 120 to 140 °C and then to 160 °C. For the 50% methane feed stream, the selectivity decreased from ~9800 (120 °C) to ~6250 (140 °C) and then to ~4800 (160 °C). The highest hydrogen selectivity (~22,200) was achieved with the 80% methane feed stream at 140 °C. It was therefore evident that the selectivity values increase with both methane concentration and current density, and decrease when the temperature is increased.

Furthermore, the temperature effect on the hydrogen selectivity of the PEM was more pronounced at higher temperatures. Compared to the hydrogen selectivity values of other membrane technologies, for example, dense metallic (>1000), dense ceramic (>1000), microporous ceramic (5–139), porous carbon (4–20), and dense polymeric (low), it is evident that in terms of H_2_/CH_4_ separation (selectivity) the EHS far exceeds its competitors [18,25].

### 4.3. Hydrogen Separation from H_2_/CO_2_ Gas Mixtures

The voltage–current characteristics of electrochemical hydrogen separation from the H_2_/CO_2_ gas mixtures, as a function of temperature, are shown in Figure 11. The curved profile of the polarization curves suggests that the electrode reaction dominates the overpotential. Here the cell showed significantly larger voltages than in the case of the performance with pure hydrogen. For the 10% CO_2_ (balance hydrogen) separation mixture, the maximum achievable current density at 160 °C was 0.16 A cm^−2^. This is a large decline from the 0.4 A cm^−2^ that was reached with pure hydrogen. Furthermore, at 120 and 140 °C, the limiting current densities decreased from 0.28 and 0.32 A cm^−2^, obtained with pure hydrogen, to 0.12 and 0.16 A cm^−2^, respectively. For the 50% CO_2_ inlet stream, the limiting current densities were 0.14, 0.14, and 0.12 A cm^−2^, respectively at 160, 140, and 120 °C. A slight performance decline, in the order of 0.02 A cm^−2^, was seen when the CO_2_ concentration was changed from 10% (Figure 11a) to 50% (Figure 11b).

The change in the overpotential characteristics can be explained by the hydrogen to impurity ratio of the inlet stream. Since the impurity (CO_2_) makes up a large portion of the gas mixture, many of the active sites are covered by impurities and, consequently, the reaction sites are ‘blocked’ by the impurities. This was also observed with the H_2_/CH_4_ gas mixtures. However, the performance decay was more severe in the presence of CO_2_ than with CH_4_, possibly caused by the electrocatalytic reduction/hydrogenation of CO_2_. Previous studies, where similar experiments were performed, reported the reduction of CO_2_ to CO by the reverse water–gas shift (RWGS) [28,35,53]:(6)$$H_{2}+CO_{2}\to CO+H_{2}O$$

Here, the adsorbed hydrogen reduces CO_2_ to CO on the electrocatalyst surface [54,55]. The formed CO strongly bind to the Pt sites, consequently causing the Pt to be poisoned [56]. Therefore, Pt–H binding is hindered [54,57]; hence, a voltage increase result.

Theoretically, the electrode tolerance to CO should increase with temperature, which is confirmed by the present work; the performance increased when the temperature was raised (Figure 11), with lower voltages seen at higher temperatures. However, no CO was detected with the in-line GC, situated downstream from the cathode outlet. Rather, trace amounts of methane were found on ppm level. This could be explained by the methanation of CO_2_ to CH_4_, believed to be driven by the presence of Co in the electrocatalyst. Co is known to have a high activity, similar to that of Ni [58,59] (Activity: Ru > Fe > Ni > Co > Mo [60]) and high selectivity (Selectivity: Ni > Co > Fe > Ru [60]) towards the methanation of CO_2_. The possible reaction network for the reduction of CO_2_ can be described by Equation (6) together with Equations (7) and (8) [61,62].
(7)$$4H_{2}+CO_{2}\to CH_{4}+2H_{2}O$$
(8)$$3H_{2}+CO\to CH_{4}+H_{2}O$$

Only one article was found on the electrochemical reduction of CO_2_ in an electrolytic cell based on polybenzimidazole (PBI) membranes doped with phosphoric acid (PA) [63]. Here, the electrocatalytic reduction of CO_2_ on Pt-Mo/C and Pt/C was examined and compared [63]. Both CO and CH_4_ were detected as CO_2_ reduction products, although the Mo addition into Pt in the Pt-Mo/C electrocatalyst significantly promoted CO formation, whilst the formation of CH_4_ was suppressed. They confirmed that CO_2_ was reduced firstly to CO by adsorbed hydrogen, and hereafter further reduced to CH_4_ by more adsorbed hydrogen.

Here, as was the case with the methane-containing mixtures, the OCV is not zero, explained by the difference in partial pressures. See Table 4. The OCVs obtained were much higher than those obtained with the H_2_/CH_4_ mixtures at the same conditions. This might be due to the fact that CH_4_ only acts as a hydrogen diluent [64], whilst CO_2_ actually affects the electrode and its performance [47].

In terms of hydrogen purity, 98–99.5% was reached with 10% CO_2_ in the feed and 96–99.5% with 50% CO_2_ in the feed stream. Therefore, a hydrogen purity of up to 99.5% can be reached with a H_2_/CO_2_ ratio of 1:1. No correlation was found between the hydrogen purity of the permeate and the temperature, whereas the hydrogen purity did increase with current density. Figure 12 shows the hydrogen selectivity values in terms of feed concentration. Reasonable separation was achieved from the 1:1 H_2_/CO_2_ mixture, with selectivities of up to ~200. This is, however, much lower than the hydrogen selectivity values obtained with the H_2_/CH_4_ gas mixtures (up to ~22,000). A similar trend was observed for both the H_2_/CH_4_ and H_2_/CO_2_ mixtures in terms of current density and impurity concentration; the hydrogen selectivity increased with current density and impurity concentration. However, the trend of the selectivity in terms of temperature differed from that of the H_2_/CH_4_ mixtures. The selectivity did not increase with a decrease in temperature; the best selectivity was achieved at 160 °C. This can be explained by the catalyst poisoning that occurs (CO poisoning)—catalyst deactivation is alleviated at higher temperatures.

Compared with other membrane technologies, the hydrogen selectivities here were lower than achieved with dense metallic and dense ceramic membranes (>1000), but higher than those obtained with microporous ceramic, porous carbon and dense polymeric membranes [18,25].

The current, voltage, and power efficiencies of the H_2_/CO_2_ gas mixtures are given in Figure 13. In general, the current efficiencies increased with current density. The lower current efficiencies at lower current densities can be attributed to hydrogen cross-over. The current efficiencies at lower current densities were lower than in the case of pure hydrogen (~70% at 0.02 A cm^−2^). In terms of voltage efficiencies, an increase was observed when the temperature was increased, but a decrease with an increase in current density. A similar trend was seen for both the pure hydrogen and the hydrogen–methane mixtures.

The power efficiencies showed a maximum value of 0.08–0.10 A cm^−2^ for both the 10 and 50% CO_2_ inlet mixtures. This depicts the optimum current density for hydrogen separation from the H_2_/CO_2_ mixtures at the respective operating conditions with regards to the power consumption. Similar to the results recorded for the methane-containing mixtures, the power efficiencies here decreased with an increase in carbon dioxide concentration and increased with temperature. Moreover, nearly Faradic flow rates were achieved in all experiments performed.

### 4.4. Hydrogen Separation from H_2_/NH_3_ Gas Mixtures

The voltage–current characteristic of the H_2_/NH_3_ inlet mixtures are presented in Figure 14. The cell performance is severely impaired in the presence of ammonia, even in very low concentrations (ppm range), when compared to the performance with pure hydrogen. The limiting current densities achieved were 0.10, 0.12, and 0.12 A cm^−2^ for 120, 140, and 160 °C, respectively, when 1500 ppm ammonia was present. When the cell was operated in the presence of 3000 ppm ammonia, the cell performance degraded completely, even with an increase in temperature. The limiting current densities in the presence of 3000 ppm NH_3_ were 0.04, 0.04, and 0.02 A cm^−2^ for 120, 140, and 160 °C, respectively. The limiting current densities seemingly indicate that the performance of the cell decreases as the temperature is raised. A reverse trend is, however, seen in the presence of 1500 ppm NH_3_. The results indicated that the performance of the cell deteriorates in the presence of ammonia (continuous exposure to ammonia) rather than with temperature, as observed in Figure 14b.

The selectivity graphs presented in Figure 15 show that very poor selectivities (<2) are achieved for feed streams containing both 1500 and 3000 ppm ammonia. Ammonia and hydrogen seem to compete with each other. In the cases where the selectivity values < 1, ammonia transport through the membrane is favored, whereas hydrogen is favored at selectivities > 1. Since hydrogen is split into its constituents, more specifically, protonated to H^+^ ions, ammonia protonation results:(9)$$NH_{3}+H^{+}\to NH_{4}^{+}$$

NH_3_^+^ is then also transported through the membrane.

The anode reaction can be depicted as follows:(10)$$NH_{3}+2H_{2}\to NH_{4}^{+}+H^{+}$$

At lower current densities, however, the reaction seems to be the following, where ammonia is favored and not hydrogen:(11)$$2NH_{3}+H_{2}\to 2NH_{4}^{+}$$

The effect of ammonia on the performance of LT-FCs has been determined, and reported in literature [65,66,67]. Some sources even report a decrease in FC catalytic activity when only 1 ppm NH_3_ is present, over a short one-week period. Their tests were performed with Nafion^®^ (Wilmington, DE, USA) membranes. The loss in activity can be attributed to the alkaline nature of ammonia. The formed NH_4_^+^ ions (Equation (11)) react with the sulphonic groups (SO_3_^−^), which are characteristic of Nafion membranes. Consequently, the water content and the conductivity of the membrane decreases, resulting in reduced FC performances [66,67]. Likewise, as for the sulphonic groups of the Nafion^®^ membranes, so too for PA used in HT membranes (usually PBI-based membranes), which results in phosphate groups (PO_4_^3−^). The performance losses observed in Figure 15 are therefore attributed to the alkaline nature of ammonia, reacting with the PA on the membrane. More specifically the formed NH_4_^+^ ions react with the PO_4_^3−^ groups, consequently resulting in a loss of conductivity. Similar to the results reported here, Vassiliev et al. [68] reports on the possible chemical incompatibility between the other organic fuels and the electrolyte in HT membranes doped with PA. More specifically, it was concluded that the probable reasons were that the catalyst deactivation and/or acid depletion within the electrode occurred.

Results of studies carried out with Nafion membranes to determine the effect of ammonia on the performance of LT-FCs have revealed that although substantial deactivation effects were observed in the presence of ammonia in the feed gas, the cell performance was recoverable, with time (in the order of days), when neat hydrogen is introduced to the anode side [67]. In the present study, however, this did not apply—the membranes were deactivated beyond recovery; for example, membrane cracks occurred (see Figure 16). Scanning electron microscopy (SEM) images of the microscopic morphological changes on the surface of a membrane and GDLs are also shown in Figure 17. Small cracks are even visible at the microscopic level.

### 4.5. Possible Future Work

As for a possible future work, one can consider using hydrocarbon-based PEM membranes [69], PFSA-based materials with various modifications to address water management challenges [70], as well as novel phosphonated materials [40].

## 5. Conclusions

For hydrogen feed, the voltage–current characteristics showed that the ohmic losses dominated the overpotential. From this, it was concluded that the membrane- and contact resistances of the cell play a major role in the efficacy of the EHS, and should, therefore, be minimised. Overall, the general performance of the cell increased with minor temperature changes (20 °C). Moreover, the current efficiencies for the pure hydrogen inlet ranged from ~70% to tlevels >95%. The current efficiency increased with current density, whilst the voltage efficiency increased with temperature and decreased with current density. This resulted in higher power efficiencies as the temperature was raised. The optimal current density with regards to power consumption was ~0.16 A cm^−2^. The highest limiting current density was 0.4 A cm^−2^, which is well below a current density of practical interest, 1 A cm^−2^.

Results revealed that hydrogen can be effectively recovered from H_2_/CH_4_ mixtures, yielding a hydrogen purity of >99.9%. From the polarization curves and selectivity graphs of the H_2_/CH_4_ mixtures, it was evident that the selectivity decreases with temperature, whilst the overall cell efficiency (power efficiency) increases, therefore the selectivity and cell efficiency behave antagonistically towards another. The highest selectivity achieved with H_2_/CH_4_ was ca. 22,000. Similar trends with regards to current, voltage, and power efficiencies were visible when compared with trends of the pure hydrogen experiments. Furthermore, performance of the cell was seen to decrease with an increase in methane concentration. Therefore, the cell performance is sensitive to the partial pressure of the hydrogen concentration in the feed.

In the H_2_/CO_2_ experiments, the Pt/Co catalyst is severely influenced by CO_2_, even at elevated temperatures. Hydrogen purities of 98–99.5% can be achieved for feed streams containing 10% CO_2_ and 96–99.5% for in the feed streams containing 50% CO_2_. Overall, poor selectivities (≤200) were reached for the CO_2_-containing streams, with poor overall cell performances due to catalyst deactivation caused by CO_2_ reduction. Similar to the H_2_/CH_4_ mixtures, the power efficiencies decreased with an increase in CO_2_ concentration and increased with temperature. The power efficiencies were, however, lower than that obtained by pure hydrogen and the CH_4_-containing mixtures.

Ammonia has a noticeable effect on the membrane—ammonia reacts with the acid embedded in the membrane. Even at very low concentrations (ppm) the membranes were affected by ammonia beyond recovery and, hence, unsuitable for use in electrochemical hydrogen separation. However, regarding the proton transport through the membrane, ammonia and hydrogen competed; ammonia transport was favored at low current densities and hydrogen transport was favored at higher current densities. The formation of NH_4_^+^ demonstrates the possibility of electrochemical transport/pumping and compression of ammonia with a PEM. In future research, alternative anion exchange membranes could be considered to possibly facilitate this reaction. 

In terms of the hydrogen selectivity, power efficiencies, and the effects on the cell components, the performance of the cell based on the different gas mixtures were in the order: H_2_/CH_4_ > H_2_/CO_2_ > H_2_/NH_3_.

## Figures and Tables

**Figure 1 membranes-11-00282-f001:**
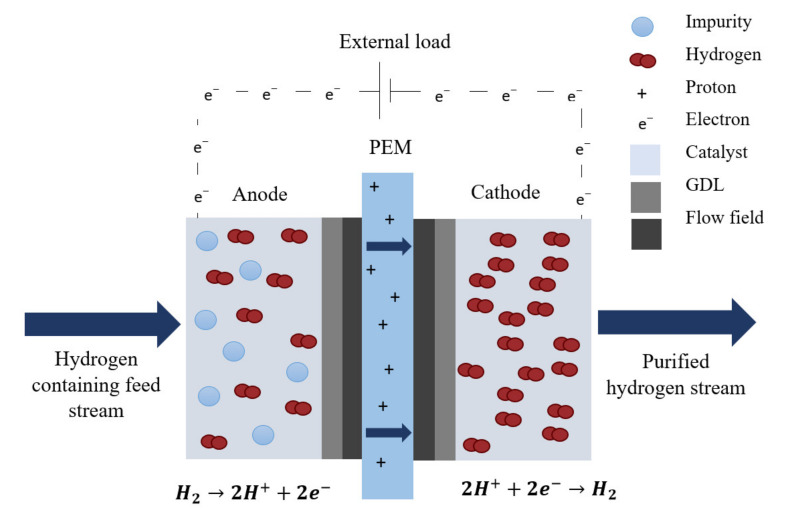
Working principal of an ideal electrochemical hydrogen separator [42,43].

**Figure 2 membranes-11-00282-f002:**
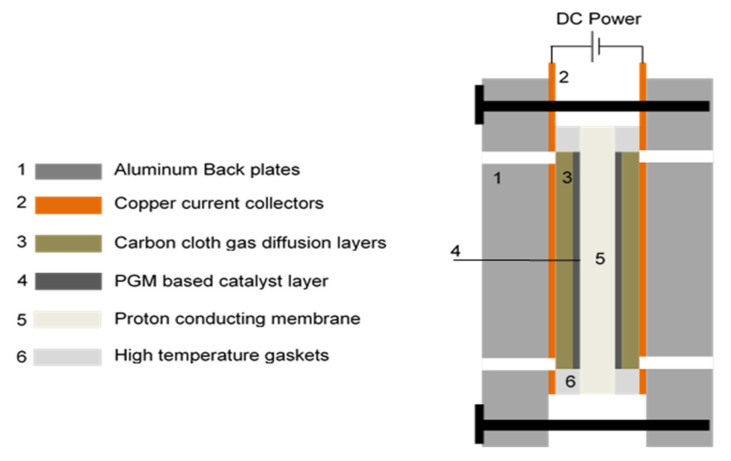
Cross section of the electrochemical cell.

**Figure 3 membranes-11-00282-f003:**
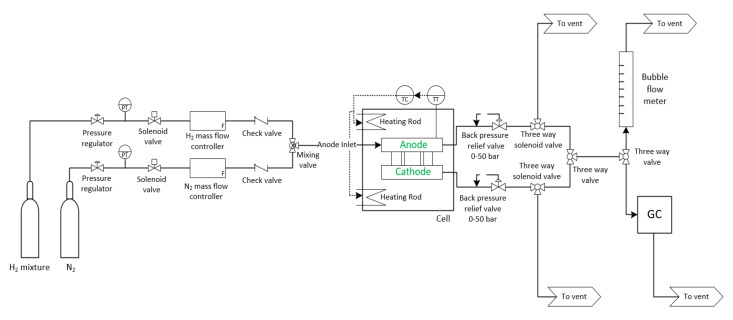
Experimental set-up used for electrochemical hydrogen separation.

**Figure 4 membranes-11-00282-f004:**
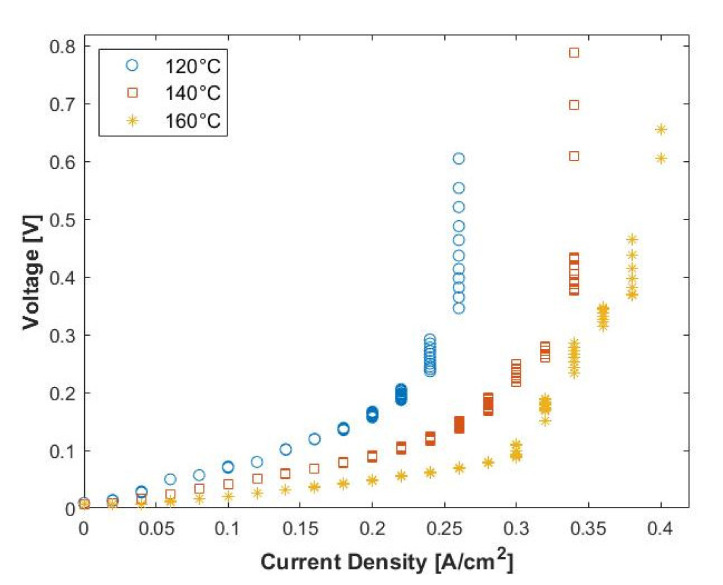
Polarization curve for pure hydrogen (100 mL_n_/min, 1 atm).

**Figure 5 membranes-11-00282-f005:**
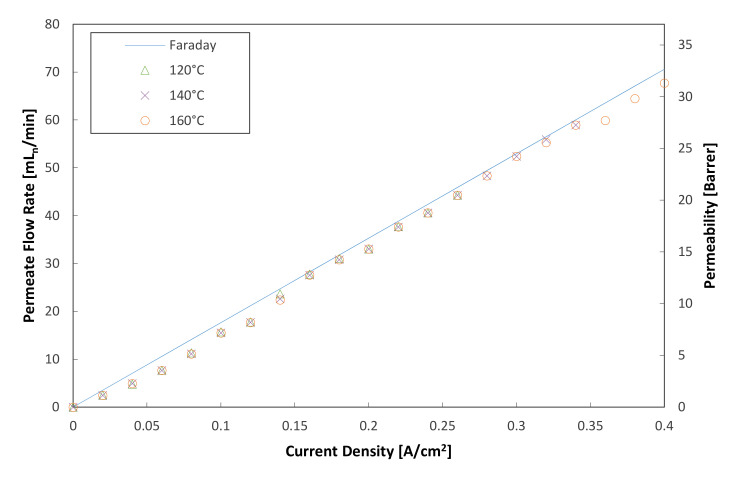
Theoretical and measured hydrogen permeate flow rates and hydrogen permeability (hydrogen flow rate = 100 mL_n_/min, 1 atm).

**Figure 6 membranes-11-00282-f006:**
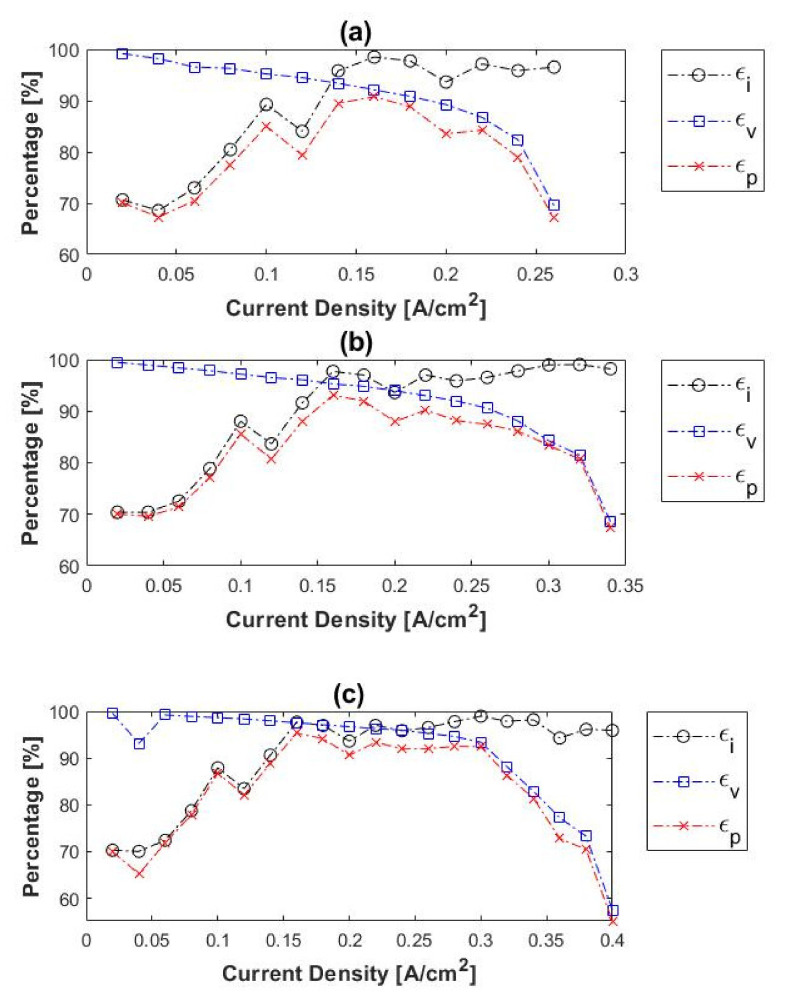
Current-(ε_i_), voltage-(ε_v_), and power (ε_p_) efficiencies as a function of current density at (**a**) 120 °C, (**b**) 140 °C and (**c**) 160 °C (hydrogen flow rate = 100 mL_n_/min, 1 atm).

**Figure 7 membranes-11-00282-f007:**
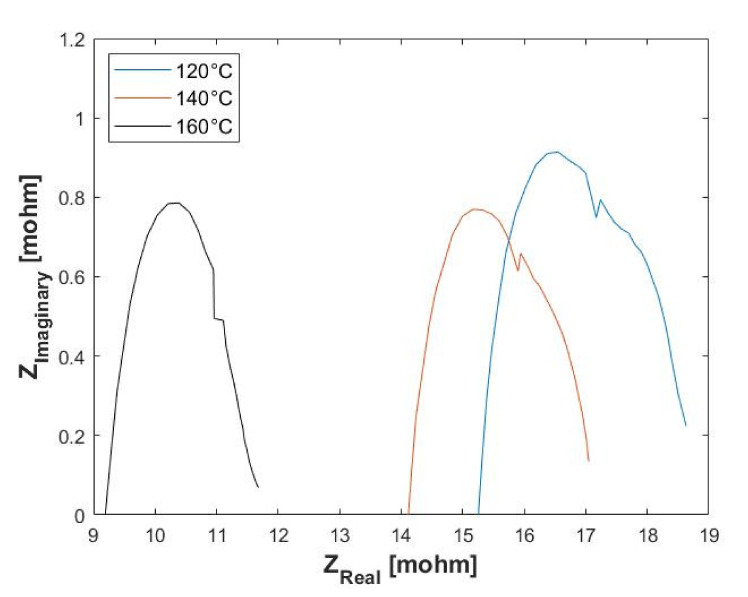
Electrochemical impedance spectroscopy (EIS) data showing the change in ohmic resistance (x-intercept) with temperature. Hydrogen flow rate = 100 mL_n_/min, 1 atm.

**Figure 8 membranes-11-00282-f008:**
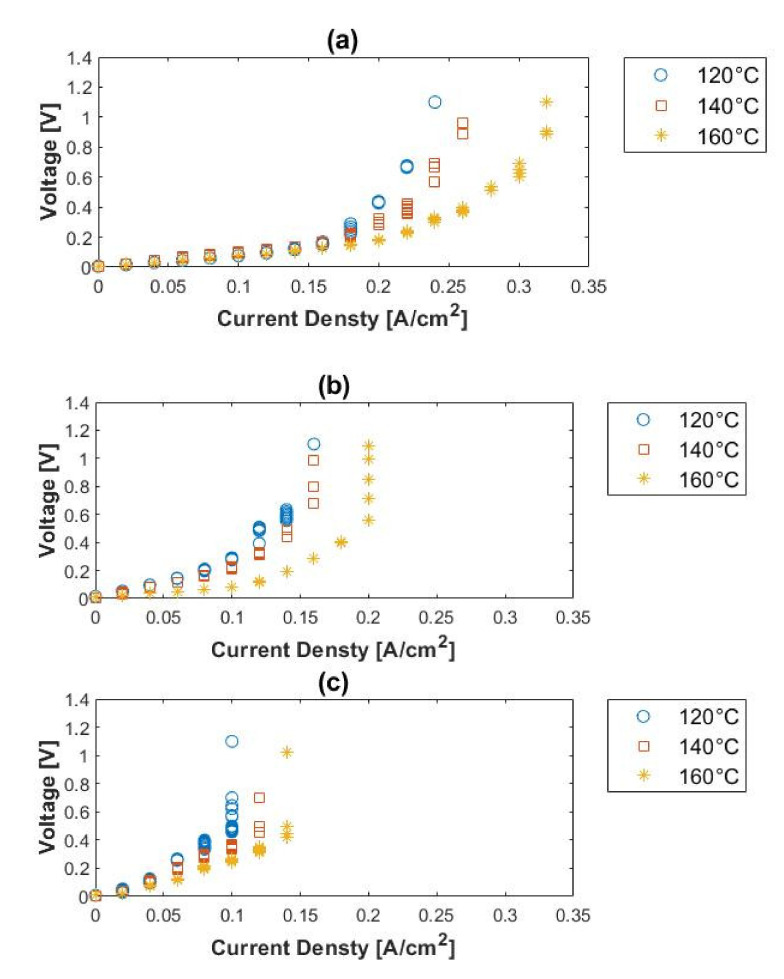
Polarization curves for (**a**) 10%, (**b**) 50%, and (**c**) 80% methane, balance hydrogen (hydrogen flow rate = 100 mL_n_/min, 1 atm).

**Figure 9 membranes-11-00282-f009:**
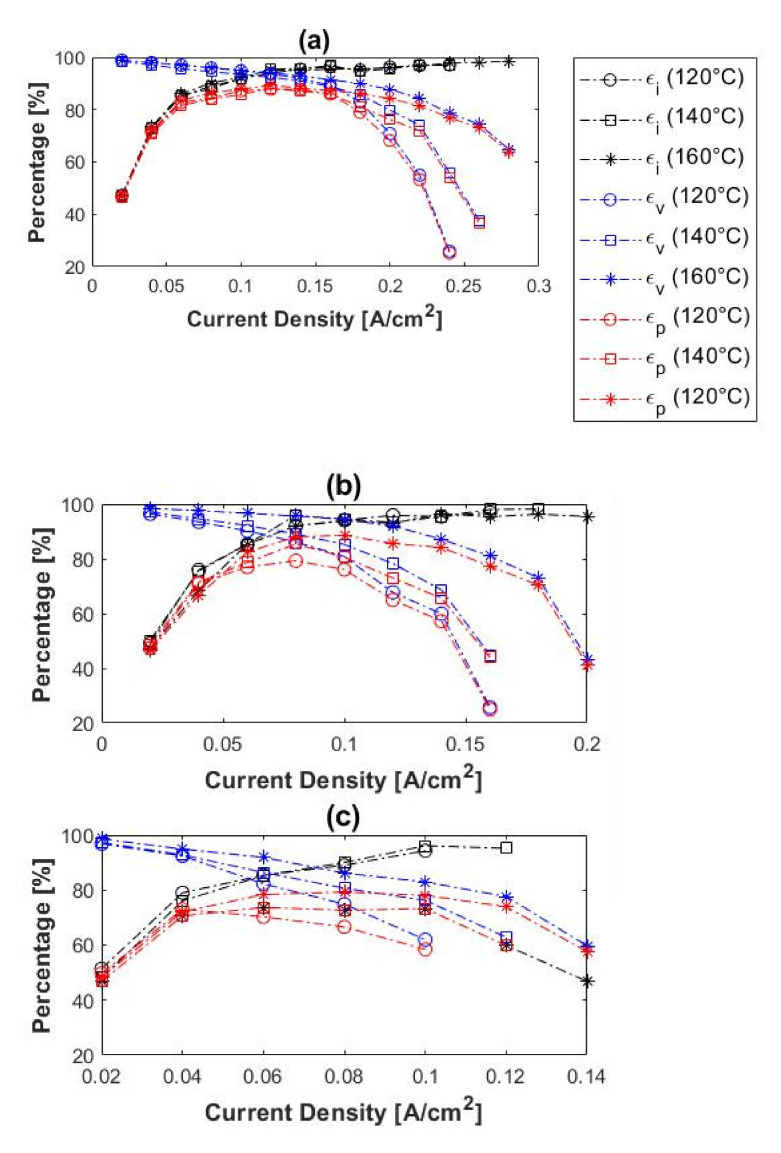
Current-(ε_i_), voltage-(ε_v_), and power (ε_p_) efficiencies as a function of current density at 120–160 °C for (**a**) 10%, (**b**) 50%, and (**c**) 80% methane, balance hydrogen (hydrogen flow rate = 100 mL_n_/min, 1 atm).

**Figure 10 membranes-11-00282-f010:**
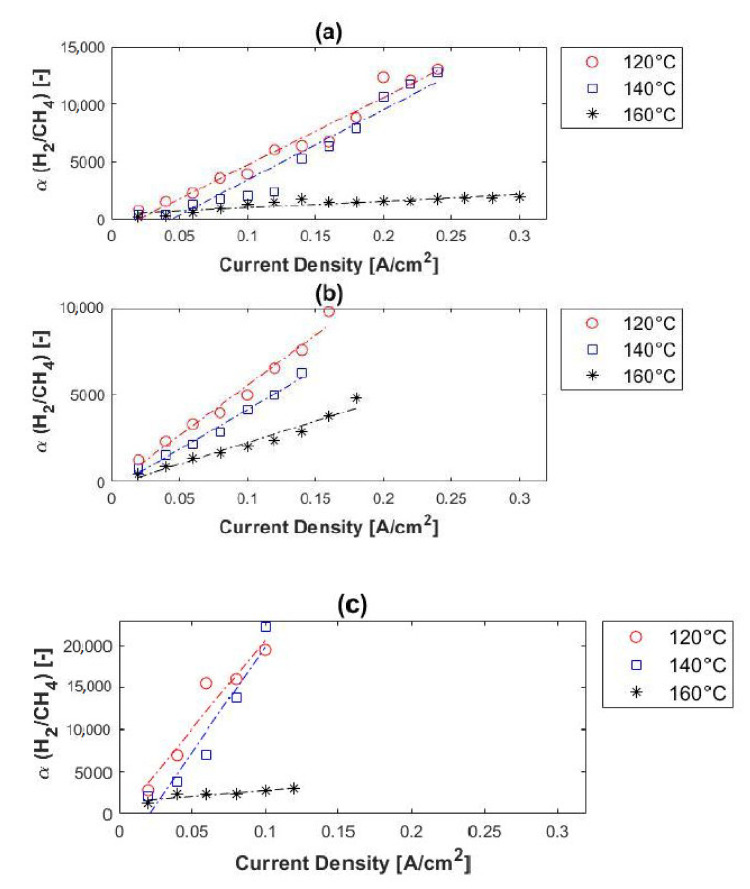
H_2_/CH_4_ selectivity, in terms of methane concentration, versus current density: (**a**) 10% (**b**) 50% and (**c**) 80% methane, balance hydrogen (hydrogen flow rate = 100 mL_n_/min, 1 atm).

**Figure 11 membranes-11-00282-f011:**
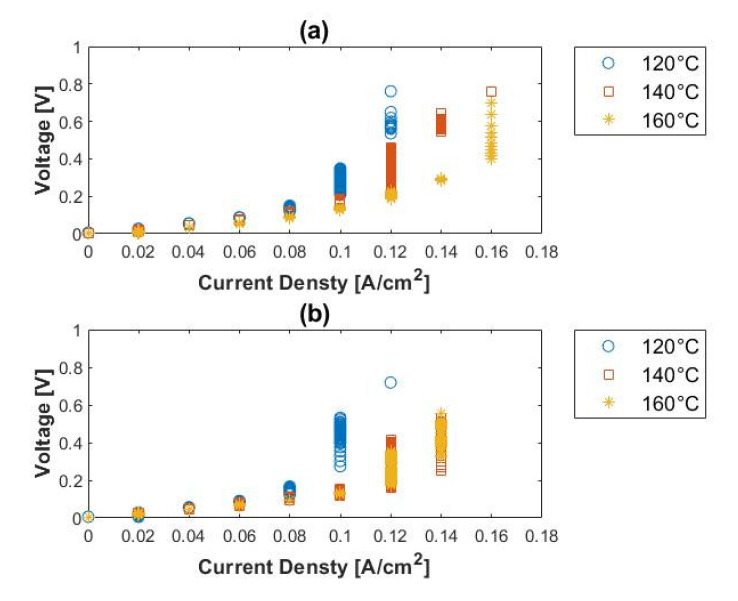
Polarization curves for (**a**) 10%, (**b**) 50% carbon dioxide, balance hydrogen (hydrogen flow rate = 100 mL_n_/min, 1 atm).

**Figure 12 membranes-11-00282-f012:**
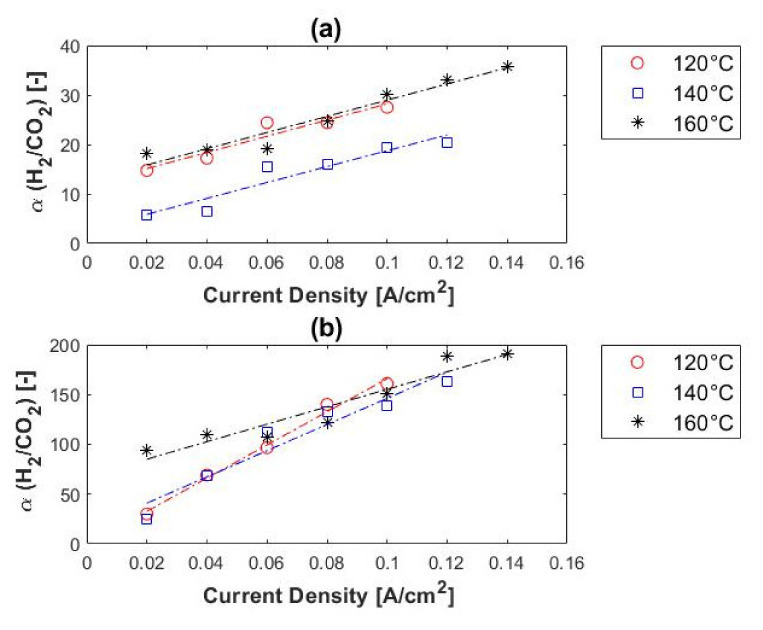
H_2_/CO_2_ selectivity, in terms of carbon dioxide concentration, versus current density: (**a**) 10% and (**b**) 50% carbon dioxide, balance hydrogen (hydrogen flow rate = 100 mL_n_/min, 1 atm).

**Figure 13 membranes-11-00282-f013:**
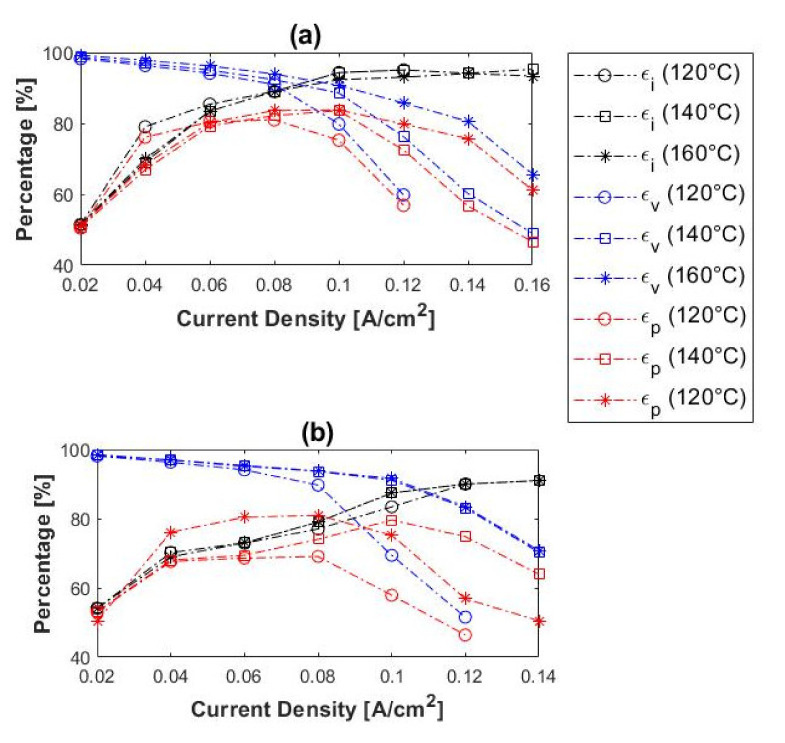
Current- (ε_i_), voltage- (ε_v_), and power (ε_p_) efficiencies as a function of current density at 120–160 °C for (**a**) 10% and (**b**) 50% carbon dioxide, balance hydrogen (hydrogen flow rate = 100 mL_n_/min, 1 atm).

**Figure 14 membranes-11-00282-f014:**
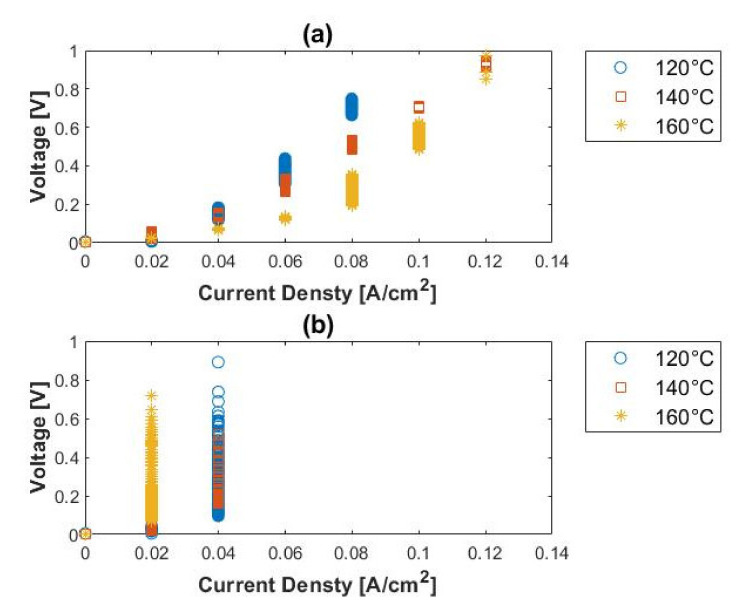
Polarization curves for (**a**) 1500 ppm, (**b**) 3000 ppm ammonia, balance hydrogen (hydrogen flow rate = 100 mL_n_/min, 1 atm).

**Figure 15 membranes-11-00282-f015:**
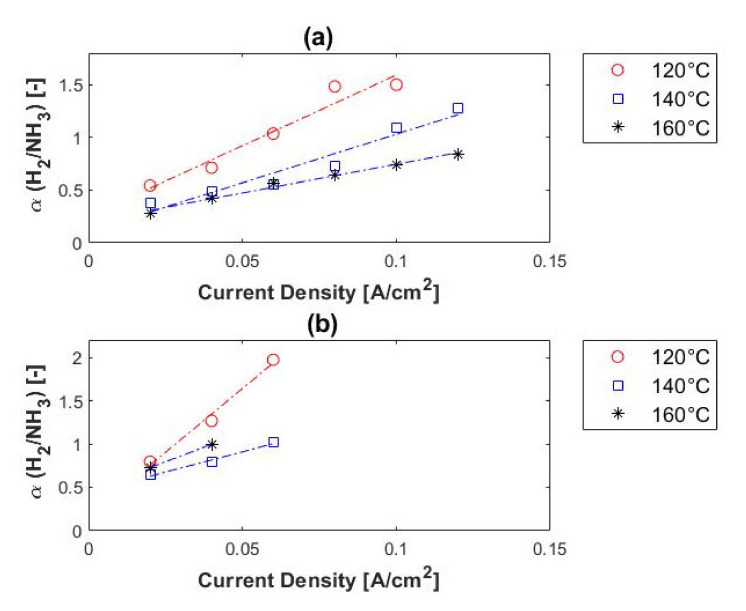
H_2_/NH_3_ selectivity, in terms of ammonia concentration, versus current density: (**a**) 1500 ppm and (**b**) 3000 ppm ammonia, balance hydrogen (hydrogen flow rate = 100 mL_n_/min, 1 atm).

**Figure 16 membranes-11-00282-f016:**
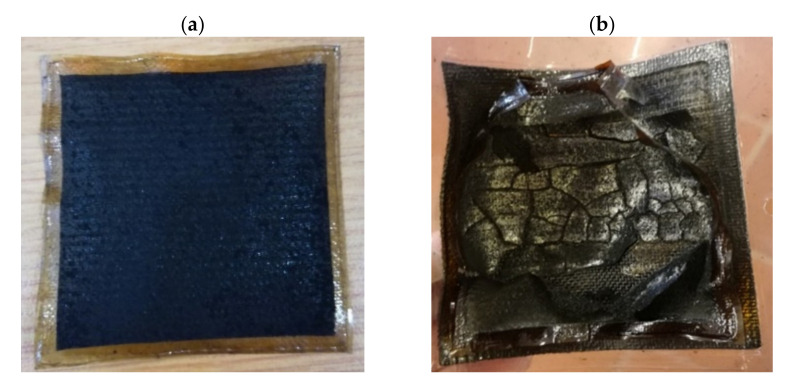
Digital images of polybenzimidazole (PBI) membranes before and after ammonia exposure: (**a**) fresh membrane (before) and (**b**) cracked membrane (after).

**Figure 17 membranes-11-00282-f017:**
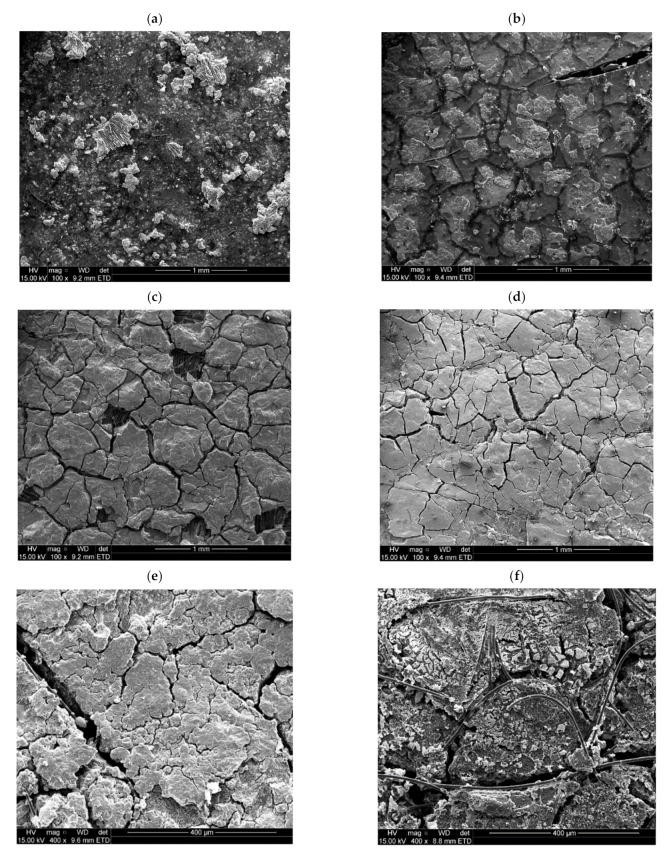
SEM images/surface morphologies of MEA components (before and after ammonia exposure): (**a**) membrane before ammonia exposure, (**b**) affected membrane after ammonia exposure, (**c**) cathode GDL (Pt catalyst) before ammonia exposure, (**d**) cathode GDL (Pt catalyst) after ammonia exposure, (**e**) anode GDL (Pt/Co catalyst) before ammonia exposure and (**f**) anode GDL (Pt/Co catalyst) after ammonia exposure (MEA: membrane electrode assembly; GDL: gas diffusion layer).

**Table 1 membranes-11-00282-t001:** Properties of different hydrogen purification processes [22,23,24].

Properties	PSA	Membranes	Cryogenic
Min. feed purity (vol.%)	>40	>25	15–80
Max. product purity (vol.%)	>99.9	>98	~97
Max. hydrogen recovery (%)	Up to 90	Up to 95	Up to 98
Inlet pressure (bar)	10–70	14–138	14–83
Outlet pressure (bar)	Similar to feed	Substantially less than feed	Similar to feed

**Table 2 membranes-11-00282-t002:** Properties of the TPS^®^-based membrane (as received from the supplier).

Membrane Type	TPS^®^-Based
Membrane thickness	60–65 μm
Catalyst used anode	Pt-Co/C
	Atomic ratio 1:1 (Pt:Co)
Catalyst used cathode	Pt/C
Total catalyst loading	1.8 mg cm^−2^
Membrane area	25 cm^2^
Temperature range	120–200 °C
Proton conductivity	8 × 10^−2^ S cm^−1^

**Table 3 membranes-11-00282-t003:** Hydrogen purity and cell voltage values as a function of temperature for the H_2_/CH_4_ feed mixtures, measured at 0.1 A cm^−2^.

Temperature	Hydrogen Purity (%)		OCV (V)	Measured Voltage (V)	Measured-OCV (V)
	Inlet	Permeate			
120 °C	20	99.98	0.027	0.564	0.537
	50	99.98	0.012	0.284	0.272
	90	99.99	0.002	0.077	0.075
140 °C	20	99.98	0.029	0.352	0.323
	50	99.98	0.012	0.219	0.207
	90	99.99	0.002	0.075	0.074
160 °C	20	99.86	0.030	0.253	0.233
	50	99.95	0.013	0.083	0.070
	90	99.99	0.002	0.074	0.072

**Table 4 membranes-11-00282-t004:** Hydrogen purity and cell voltage values as a function of temperature for the H_2_/CO_2_ feed mixtures, measured at 0.1 A cm^−2^.

Temperature	Hydrogen Purity (%)		OCV (V)	Measured Voltage (V)	Measured-OCV (V)
	Inlet (Anode)	Permeate (Cathode)			
120 °C	50	99.39	0.012	0.454	0.442
	90	99.56	0.002	0.300	0.298
140 °C	50	99.29	0.012	0.132	0.120
	90	99.44	0.002	0.170	0.168
160 °C	50	99.34	0.013	0.125	0.112
	90	99.63	0.002	0.138	0.136

## Data Availability

Not applicable.

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
