# Peer review of "Hydrogen Separation and Purification from Various Gas Mixtures by Means of Electrochemical Membrane Technology in the Temperature Range 100–160 °C"

_membranes, 2021, doi:10.3390/membranes11040282_

Round 1

Reviewer 1 Report

The manuscript titled “Hydrogen separation and purification from various gas mixtures by means of electrochemical membrane technology in the temperature range 100-160°C”by Vermaak et al reported that separation and purification of hydrogen gas from H2/CO2, H2/CH4 and H2/NH3 gas mixture by using proton exchange membrane . Authors studied effect of impurity concentration and hydrogen separation at high operating temperature at 100-160°C. The data presented in the manuscript are adequate to prove the results presented by the authors. Overall, the manuscript is sound. Therefore, based on the merits, I strongly recommend this manuscript for publication in Membranes. However, before the publication the following comments should be thoroughly revised.

General comment.

  1. Introduction section is too long and authors should rewrite introduction part with important of proton exchange membrane towards hydrogen separation and purifications and use recent literature report.
  2. How the hydrogen separated from mixture of gases by using proton exchange membrane? If possible, author can explain by using schematic diagram.
  3. How did find out the purity of H2?. Have you used any chromatography analysis after separated H2 gas? Explain.
  4. The conclusion part should be more elaborate.
  5. Some of the important references are need to cite about the proton exchange membrane in revised introduction part. DOI: 10.1002/er.4494, 1021/acssuschemeng.9b01757.

Reviewer 2 Report

The authors describe the electrochemical hydrogen pump characteristics of a high temperature membrane electrode assembly. While I am impressed by the quality and experimental design of the experiments, I am concerned that very little discussion is centered on membrane technology. My comments are listed below:

  1. Why was TPS membrane and advent electrodes (PtCo/C alloy anode, Pt/C cathode) selected when electrochemical pump systems utilize Pt/C on both anode and cathode?
  2. The TPS membrane contains phosphoric acid. How is the doping level? What was the thickness of the membrane? How was the MEA preconditioned, prior to measurements?
  3. As mentioned, the TPS membrane contains phosphoric acid. While this is an not issue when flowing pure hydrogen, flowing H2/CO2 and H2/NH3, CO2 and NH3 can contaminate the phosphoric acid in the membrane. Please cite https://doi.org/10.1016/j.jpowsour.2019.05.062 by Aili et. al, which discusses the contamination of high temperature membranes by organic fuels at elevated temperature. While this is mentioned on page 17, this discussion should a major component of this section.
  4. The authors discus the change in overpotential in the H2/CO2 experiments due to electrocatalytic reduction/hydrogenation of CO2. While I agree with this point, an EIS measurement at various potentials should be conducted to see the membrane resistance increased. As CO2 electrocatalytic reduction/hydrogenation also may reduce the conductivity of phosphoric acid due to CO* absorption.
  5. There have been other advancements in high temperature membrane technology other than TPS and PBI, most notably by Y.S. Kim and A.S. Lee (Nature Materials 2021, https://doi.org/10.1038/s41563-020-00841-z and Nature Energy 2016, https://doi.org/10.1038/nenergy.2016.120, Journal of Materials Chemistry A 2019 1039/C9TA01756A, Journal of Physical Chem B 2020 https://doi.org/10.1021/acsapm.0c01405). These works should be cited as more recent developments in HT-PEM technology.
